# Modified Technogenic Asphaltenes as Enhancers of the Thermal Conductivity of Paraffin

**DOI:** 10.3390/molecules28030949

**Published:** 2023-01-18

**Authors:** Svetlana N. Gorbacheva, Yulia Yu. Borisova, Veronika V. Makarova, Sergey V. Antonov, Dmitry N. Borisov, Makhmut R. Yakubov

**Affiliations:** 1Institute of Macromolecular Compounds, Russian Academy of Sciences, Bolshoi Prospect V.O. 31, 199004 St. Petersburg, Russia; 2A.V. Topchiev Institute of Petrochemical Synthesis, Russian Academy of Sciences, Leninsky Pr. 29, 119991 Moscow, Russia; 3Arbuzov Institute of Organic and Physical Chemistry, FRC Kazan Scientific Center, Russian Academy of Sciences, Arbuzov Str. 8, 420088 Kazan, Russia

**Keywords:** phase change material, heat conductivity, paraffin composites, ethylene tar, technogenic asphaltenes, chemical modification

## Abstract

The low thermal conductivity of paraffin and other organic phase change materials limits their use in thermal energy storage devices. The introduction of components with a high thermal conductivity such as graphene into these materials leads to an increase in their thermal conductivity. In this work, we studied the use of inexpensive carbon fillers containing a polycyclic aromatic core, due to them having a structural similarity with graphene, to increase the thermal conductivity of paraffin. As such fillers, technogenic asphaltenes isolated from ethylene tar and their modified derivatives were used. It is shown that the optimal concentration of carbon fillers in the paraffin composite, which contributes to the formation of a structural framework and resistance to sedimentation, is 5 and 30 wt. %, while intermediate concentrations are ineffective, apparently due to the formation of large aggregates, the concentration of which is insufficient to form a strong framework. It has been found that the addition of asphaltenes modified with ammonium persulfate in acetic acid significantly increases the thermal conductivity of paraffin by up to 72%.

## 1. Introduction

It is known that organic phase change materials (PCM) are capable of accumulating and releasing heat due to phase transitions and can be used to accumulate thermal energy [1]. Paraffin-based systems are widely used as such devices, since they have a high energy storage density; they are also characterized by high thermal and chemical stability, they are non-toxic, and also have a low cost [2]. The disadvantage of such systems is the low rate of heat transfer, which leads to a decrease in the performance of heat accumulators and restricts their use [3]. The introduction of substances with a high thermal conductivity, such as graphene, into these systems improves the thermal properties of PCMs. Among them, metals [4], metal oxides [5], and boron nitride [6,7] could be mentioned. Carbon-based substances, such as graphene, are among the most popular additives due to their relatively high thermal conductivity [8,9,10]. In this case, an effective increase in the thermal conductivity of PCMs is associated with the presence of the π–π interaction between planar polycyclic fragments, which promotes heat transfer [11,12,13].

Although paraffin-based PCMs containing graphene particles or carbon nanotubes exhibit higher thermal conductivity enhancement [14,15], the high cost of these carbon nanofillers restricts their application [16]. On the other hand, carbon black and graphite are much cheaper and can be used to enhance the thermal conductivity of PCMs. Wu et al. [17] reported an increase in thermal conductivity in octadecane containing carbon black nanopowder modified with oleic acid. An improvement in the thermal conductivity of paraffin wax filled with expanded flexible graphite with different densities was shown in [18]. However, in most cases, inexpensive carbon materials require pretreatment and modification to be used as PCM thermal conductivity enhancers.

Glova et al. [19], using computer simulations, have shown that other inexpensive carbon substances, petroleum asphaltenes, can increase the thermal conductivity of liquid paraffin, provided that peripheral alkane groups are removed from the polyaromatic core. In a previous work [20], we studied the properties of paraffin/petroleum asphaltene composites, but the modification of asphaltenes with sulfuric acid and oleum led only to a slight increase in the thermal conductivity of the PCMs.

As an alternative to petroleum asphaltenes, their analogues, technogenic asphaltenes, which were isolated from ethylene tar (ET), a waste product of ethylene production by pyrolysis of hydrocarbon raw materials [21,22], are considered in this paper. Currently, ET is being investigated as a raw material for the synthesis of pitch precursors for various carbon materials, including general purpose carbon fiber [22,23,24] and carbon nanofiber [25]. However, on an industrial scale, ET is used mainly for the production of carbon black or as a component of boiler fuel. It should be noted that ET is a mixture of various compounds, predominantly aromatic [26,27], and is usually characterized by the content of saturated, aromatic, and polar compounds according to the IP 469 method and technogenic asphaltenes, a residue insoluble in heptane according to the IP 143 method. Thus, ET has great potential for the production of various carbon materials and can serve as a valuable source for the extraction of polycyclic aromatic compounds.

In this work, we compared the structural similarity and difference between petroleum and technogenic asphaltenes. For technogenic asphaltenes, a number of features of the composition and properties have been identified, which allow us to consider them as a promising thermal conductivity enhancer for obtaining paraffin-based PCMs. The structure of technogenic asphaltenes (ET-A) was modified by treatment with ammonium persulfate in sulfuric acid (ET-Ox1) and acetic acid (ET-Ox2) media.

For composites of paraffin with technogenic asphaltenes and their modified derivatives, the concentrations of the fillers were determined, at the point at which they possess sedimentation stability while the crystallinity of the systems was not altered significantly. The values of the thermal conductivity for composites of paraffin with technogenic asphaltene were also determined.

## 2. Results and Discussion

### 2.1. Characteristics of Technogenic Asphaltenes and Their Modified Derivatives

It should be noted that, in comparison with petroleum asphaltenes Petroleum-A, technogenic asphaltenes ET-A are characterized by a lower molecular weight and polydispersity (Appendix A), contain almost no heteroatoms (Table 1), but also have polycyclic aromatic nuclei at the heart of the molecular architecture.

According to the Matrix Assisted Laser Desorption/Ionization (MALDI) spectrum (Appendix A), ET-A asphaltenes are characterized by a molecular weight distribution from 100 to 1300 m/z, with a peak molecular weight of M_max_ = 550 m/z. For comparison, it is worth pointing out that the molecular weight of ovalene with 10 aromatic rings is 398 g/mol, and that of perylene with five aromatic rings is 252 g/mol. Probably, technogenic asphaltenes are represented by oligomeric forms (mainly dimeric) of aromatic structural units obtained as a result of condensation at the pyrolysis of mixed gasoline. Thus, the predominant aromatic compounds previously identified by us in distillate fractions (up to 350 ℃) of ET [26] were derivatives of phenanthrene, biphenyl, naphthalene, fluorine, and pyrene. It can be assumed that the heavy residue of ET (asphaltenes ET-A) is a mixture of the oligomeric forms of these aromatic compounds, as described in [28].

In the infrared (IR) spectrum of ET-A (Figure 1a), there are intense bands of out-of-plane C-H bending vibrations of the aromatic region (3046 and 3024 cm^−1^, 964–695 cm^−1^) similar to the C-H vibrations of the ring in the same regions of ovalene’s spectrum (Figure 1b). There are also intense bands of C-H stretching vibrations of the methyl and methylene groups (2845–2975 cm^−1^, 1453 and 1376 cm^−1^), as in petroleum asphaltenes.

Studies carried out using a computer simulation of the atomic scale in [19] showed that a large number of peripheral alkyl substituents and a significant polydispersity of petroleum asphaltenes prevent the formation of extended ordered aggregates which could have been capable of heat transfer due to the π–π interaction between the planar polycyclic nuclei. Taking into account that ET-A asphaltenes also have alkyl substituents at polycyclic cores, it was necessary to find methods for modifying their structure.

Thus, the action of strong oxidizing agents on petroleum asphaltenes leads to the removal of aliphatic substituents along the perimeter of the polycondensed aromatic nucleus with the formation of carboxyl groups, which is accompanied by the compaction of the spatial structure [20,29,30]. These approaches were extended to technogenic asphaltenes ET-A, which were modified by treatment with ammonium persulfate in sulfuric acid (ET-Ox1) and acetic acid (ET-Ox2).

Indeed, for modified asphaltenes one can observe growth in the oxygen (in the case of the ET-Ox1 and ET-Ox2 products) and sulfur (in the case of the ET-Ox1 product) content (Table 1). According to XPS data, oxygen is found in the C–O-C and O-C=O groups, while sulfur in the case of the ET-Ox1 product exists in the form of R-SO_3_H sulfonates (Appendix A). The XPS spectra of sulfur of ET-Ox1 are doublets of 2p3/2-2p1/2 peaks (spin-orbit splitting) with an intensity ratio of 2/1 and a splitting of 1.18 eV (Appendix A) [31,32]. The XPS spectrum of ET-Ox1 also contains two nitrogen peaks (N1s 400.4 and 402.0 eV), which can be attributed to pyrolytic and quaternized nitrogen (Appendix A) [33,34], probably obtained as a result of the introduction of ammonium persulfate molecules and the subsequent recombinations of the polyaromatic structures.

IR spectroscopy (Table 2) also confirms an increase in oxidation (content of C=O and C-O groups) and the sulfur content (content of sulfonate RSO_3_H and sulfoxide S-O groups) in the modified products. The value of the aromaticity coefficient (C=C/CH_3_+CH_2_) also increases and the value of the aliphatic coefficient (CH_2_/C=C) decreases after modification.

The action of a strong oxidizing agent such as ammonium persulfate on technogenic asphaltenes leads, first of all, to oxidation and the introduction of oxygen with the formation of the carboxyl and ether groups and, in the case of the ET-Ox1 product, also to the formation of sulfonic acid groups. At the same time, aliphatic substituents are removed and the polycyclic aromatic structure of the molecules is densified.

### 2.2. Physicochemical Properties of Paraffin Composition Modified with Asphaltenes

#### 2.2.1. Rheological Properties

When discussing the various properties of the paraffin composites, one should keep in mind their sedimentation stability. Filler sedimentation may lead not only to inhomogeneous distribution, but also to significant changes in the properties of the whole composite. The sedimentation stability of a suspension may be provided by the yield stress [35,36,37].

It is known that paraffin melt is a Newtonian liquid: its viscosity is practically independent of the shear stress [38]. The introduction to paraffin of technogenic asphaltenes ET-A in the amount of 5 wt. % leads to the appearance of a pronounced yield stress in the systems. In the dependences of viscosity on shear stress (Figure 2a), the yield stress looks like an almost vertical region of constant stress. This may be due to the formation of a certain structure (network of contacts) by the filler particles. The strength of this network prevents particles from sedimentation. By increasing the shear stress (or shear rate), this network may break, which can be an explanation of the reverse slope of viscosity vs. the shear stress curves after overcoming the yield stress (marked areas in Figure 2a).

However, by increasing filler concentration up to 10 %, the yield stress value drops significantly. This may be explained by the increase in the probability of the formation of large agglomerates of filler particles, which leads to a decrease in the number of agglomerate particles in the system and to the lower strength of the structural framework [39]. At an ET-A concentration of 30 wt. %, the situation changes again—the yield stress recovers—which may mean that the concentration of asphaltenes becomes sufficient for the agglomerates to be able to form their rather strong structural framework.

The storage G’ and loss G’’ moduli for compositions with 5 and 30% ET-A (Figure 2b) depend very weakly on frequency, with the storage modulus exceeding the loss modulus over the entire frequency range studied. This behavior is typical for gels. In contrast, for a system with 10% ET-A above 1 rad/s, the storage and loss moduli increase with the frequency, with a crossover point where the storage modulus dominates the loss modulus. Some decreases in the modulus with frequency in the low-frequency region can be associated with the process of the destruction of a weak network. Thus, the dynamic test data correlate with the dependences of viscosity on shear stress.

A similar change in the yield strength with filler concentration is observed for composites with asphaltenes ET-OX1 (Figure 3a). The dependences of storage and loss moduli on frequency (Figure 3b), however, are weak for all systems with these types of asphaltenes, and G’ > G’’ for all composites in the entire frequency range. It should be noted that the highest modulus values are typical for compositions with 5 and 30 wt% filler, while for intermediate concentrations (10 and 20 wt. %), they are significantly lower.

For systems with ET-Ox2 asphaltenes, we also observe the deterioration of yield stress and moduli values with the increasing of the filler concentration from 5 to 20 wt% (Figure 4). The system with 20% ET-Ox2 demonstrates the same pattern of frequency dependences of the moduli as the one with 10 wt% of ET-A: there is a drop in the low-frequency range with a growth at higher frequencies, which indicates a very weak network.

The concentration dependence of the yield stress for the discussed systems is shown in Figure 5. It can be concluded from it that at asphaltenes concentrations of 5 and 30 wt%, their particles or aggregates form a stable structural network capable of preventing their sedimentation in the paraffin melt, while at intermediate concentrations, the network is much weaker.

#### 2.2.2. Structure of Composites

According to optical microscopy data, ET-A asphaltenes form spherical particles of about 1 µm in size, assembling into agglomerates of up to 30 µm in size. In dispersions with paraffin, the largest size of the agglomerates is 50 µm (Figure 6a). Asphaltenes ET-Ox1 and ET-Ox2 are particles of an uneven shape with a size of about 1 µm, forming agglomerates of up to 50 and 30 µm in size, respectively. In composites, the size of the agglomerates reaches 70 and 50 µm (Figure 6b,c). The larger size of the agglomerates in composites with ET-Ox1-modified asphaltenes can be associated with their modification, which enhances the interaction between their molecules.

According to the results of the X-ray diffraction analysis, the introduction of asphaltenes of various natures into paraffin has practically no effect on its crystal lattice. Compared to the diffraction pattern of pure paraffin, the diffraction patterns of composites containing asphaltenes lose the peak characteristic of paraffin in the region of 10° in 2θ (Figure 7), and at 15–20°, an insignificant amorphous halo is observed, indicating some amorphization of the structure. One can also note a decrease in the intensity of the peaks in the region of 35–45°. Thus, the data obtained indicate a decrease in the degree of crystallinity of the systems upon the introduction of asphaltenes; however, the crystal structure characteristic of paraffin is retained. 

The degree of the crystallinity of systems is linearly related with the amount of heat that will be absorbed or released during the phase transition—the greater the degree of crystallinity, the more heat the system is able to accumulate [40]. The degree of crystallinity of the samples was estimated as the ratio of the heat of fusion of the material experimentally determined by differential scanning calorimetry (DSC) to the heat of fusion of eicosan (244.2 J/g) [41], corrected for the mass fraction of paraffin in a specific system. The introduction of additives in paraffin in the amount of 5–10 wt. % leads to a decrease in paraffin crystallinity by almost 20% (Figure 8). A further increase in the filler concentration slightly improves the situation in the case of ET-Ox1 asphaltenes, stabilizes the values for ET-A, and decreases when filling with ET-Ox2 asphaltenes. Probably, this behavior is associated with the formation of agglomerates by filler particles, which interfere with the packing of paraffin chains.

#### 2.2.3. Thermal Conductivity

The rate of the accumulation and release of thermal energy by a thermal accumulator depends on the thermal conductivity of the composite material [42]. The introduction of technogenic asphaltenes into paraffin leads to a noticeable increase in its thermal conductivity with the growth of the filler concentration (Figure 9). In the case of ET-A and ET-Ox1 asphaltenes, the thermal conductivity values of the composites practically coincide, despite the lower initial value of the thermal conductivity of ET-A asphaltenes compared to paraffin. In this case, the greatest increase in thermal conductivity is 18% which is achieved at a 30 wt. % filler content. In the case of ET-Ox2 asphaltenes, the results stand out noticeably. The thermal conductivity of asphaltenes ET-Ox2 turns out to be higher than the thermal conductivity of paraffin by 58%, and their addition to paraffin in an amount as small as 5 wt. % significantly increases its thermal conductivity, and the maximum increase (by 72%) is achieved with a filler content of 30 wt. %.

This behavior should be related to the modification of asphaltenes and opens up the possibility of their successful use as a filler to increase the thermal conductivity of paraffin.

## 3. Materials and Methods

### 3.1. Materials

Highly purified paraffin grade P-2 (Lukoil, Russia), which is a mixture of solid hydrocarbons with a predominantly normal structure and a melting point of 52–58 °C, was used as a phase change material.

ET, a byproduct of pyrolysis of mixed gasoline in the production of ethylene, was provided by PJSC Nizhnekamskneftekhim (Russia). The complete characterization of ET (fractional composition, group hydrocarbon composition of distillate fractions, etc.) is presented in [26]. Technogenic asphaltenes, designated as “ET-A”, and petroleum asphaltenes, designated as “Petroleum-A”, were isolated by the IP 143 method from ET and crude oil from the Ashalchinskoye field (Russia), respectively. The yield of ET-A asphaltenes from ET was 14 wt%, whereas the weight fraction of Petroleum-A asphaltenes in the crude oil was 7.1 wt%. All solvents and reagents used in this study were obtained from Sigma-Aldrich and used as received.

#### 3.1.1. Modification of ET-A Asphaltenes with Ammonium Persulfate in Sulfuric Acid

A total of 20 g of ammonium persulfate was mixed with 60 mL of concentrated sulfuric acid (98 wt% concentration), and then 2 g of ET-A asphaltenes were added to the resulting mixture by stirring in small portions over 1 h. A strong exothermic effect and foaming were observed. After adding the last portion of asphaltenes, stirring was continued for 1 h, after which 50 mL of distilled water was slowly added and kept under stirring for 2 h more. Next, the reaction mass was poured into 500 mL of distilled water, and the black precipitate was filtered off, which was washed with water until neutral and dried under vacuum. The yield of the product, hereinafter referred to as “ET-Ox1”, was 88 wt%.

#### 3.1.2. Modification of ET-A Asphaltenes with Ammonium Persulfate in Acetic Acid

A total of 1 g of ET-A asphaltenes was dissolved in 30 mL of methylene chloride. Then, 60 mL of glacial acetic acid followed by 20 g of ammonium persulfate (in small parts) was added, while stirring, to the solution. Then, 0.1 mL of sulfuric acid (98%) was added as a catalyst. The reaction mass was stirred for another 3 h, after which 10 mL of distilled water was added and stirred for another hour. After that, the reaction mass was kept for 12 h, and then poured into 500 mL of distilled water, and the aqueous layer was separated from the black organic layer. The organic layer was evaporated from methylene chloride and the residue was dried under vacuum. The yield of the product, hereinafter referred to as “ET-Ox2”, was 98 wt %.

### 3.2. Preparation of Composites

Technogenic asphaltenes and their modified derivatives were ground in an IKA A11 basic mill at 28,000 rpm for 10 min. Composite samples were prepared by introducing 30 wt% of the filler obtained by grinding into the molten paraffin at 80 °C with constant stirring using an overhead stirrer, Heidolph RZR 2051, at 500 rpm. To obtain samples with a lower filler concentration, the concentrated mixtures were diluted step by step with paraffin to the desired concentration. Three series of samples were obtained in this way: composites filled with technogenic asphaltenes ET-A and modified asphaltenes ET-Ox1 and ET-Ox2. The filler concentration was 30, 20, 10, and 5 wt % for each system.

### 3.3. Methods

#### 3.3.1. MALDI

Matrix-assisted laser desorption/ionization mass spectra were obtained by UltraFlex III TOF/TOF mass-spectrometer (Bruker Daltonik GmbH, Bremen, Germany) in a linear mode with Nd:YAG laser (λ = 355 nm). The spectra were obtained with 25 kV of accelerating voltage and 30 ns of acceleration delay. The data were processed by using FlexAnalysis 3.0 software (Bruker Daltonik GmbH, Bremen, Germany). Additionally, 1,8,9-Trihydroxyanthracene was used as a matrix. The metal target MTP AnchorChipTM was used. In addition, 0.5 µL of 1% solutions of the matrix and the sample in toluene were successfully spotted on the target and evaporated. Spectra were collected for all spots using 250 shots per spectrum. Positively charged ions were analyzed.

#### 3.3.2. XPS

X-ray photoelectron spectroscopy was performed on a PHI 5000 Versa Probe II spectrometer (ULVAC-PHI, Inc., USA). The excitation source was monochromatized Al Kα radiation (hν = 1486.6 eV), with power of 50 W and beam diameter of 200 µm. We used a charge neutralization system because a slight electrostatic charge was observed, which distorted the spectra. Atomic concentrations were determined from survey spectra using the relative elemental sensitivity factors of the C1s, O1s, N1s, and S2p lines. The binding energies, Eb, of photoelectronic lines were determined from high-resolution spectra, taken at an analyzer transmission energy of 23.5 eV and a collection density of 0.2 eV/step. The binding energy scale, Eb, was calibrated for Au4f—83.96 eV—and Cu2p3—932.62 eV. The Eb scale was corrected for the first peak in the C1s spectrum (with the lowest Eb value of 284.7 eV), since the samples contain a large number of carbon atoms in aromatic fragments—the presence in the C1s spectrum with Eb peak in the region of 291 eV, the so-called shake-up satellite representing sp2 carbon bonds.

#### 3.3.3. Elemental Analysis

Elemental composition of samples was determined using a CHNS analyzer Vario Macro cube (Elementar Analysensysteme GmbH, Germany).

#### 3.3.4. IR Spectroscopy

IR spectra of the compounds were recorded in the range of 4000–400 cm^−1^ on a Vector-22 IR-Fourier spectrometer (Bruker, Germany) with an optical resolution of 4 cm^−1^. Samples were prepared in tablets with KBr (Acros Organics 206391000). To study the structural and group composition of samples, spectral coefficients were calculated: CH_2_/C=C—aliphaticity (D_1450_/D_1600_), C=C/CH_2_+CH_3_—aromaticity (D_1600_/D_1450+1380_), C=O/C=C—degree of oxidation I (D_1720_/D_1600_), C-O-C/C=C—degree of oxidation II (D_1230_/D_1600_), RSO_3_H—degree of sulfonation (D_1030_/D_1600_), and SO = D_1160_/D_1600_—(degree of sulfoxidation). The spectra were processed and analyzed using OPUS Version 6.5 software (Bruker Optik GmbH, Germany).

#### 3.3.5. Optical Microscopy

The study of the morphology of the obtained samples was carried out using optical microscopy. A Biomed microscope with 10x magnification and a digital video camera was used to obtain images of filled samples.

#### 3.3.6. Rheological Properties

The rheological properties of the compositions obtained were studied on a DHR-2 rotary rheometer (TA Instruments, USA) using a measuring cone-plane unit (the cone diameter was 25 mm, the angle between the conical surface and the plate was 2°). The flow curves of the samples were obtained at 80 °C with a stepwise increase of the shear rate from 10^−3^ to 1000 s^−1^.

#### 3.3.7. X-ray Diffraction

The effect of filler introduction on the degree of crystallinity of paraffin was studied by X-ray diffraction on a Rigaku Rotaflex D/max-RC instrument (Rigaku, Japan, Japan) using CuKα radiation with a wavelength of λ = 0.154 nm. Diffractograms were obtained in the range of 3–70° in 2θ with a step of 0.04° and a recording rate of 4°/min.

#### 3.3.8. Thermophysical Properties

The amount of heat released or absorbed by pure paraffin and its composites filled with asphaltenes as a result of the phase transitions was measured by DSC on the MDSC 2920 calorimeter (TA Instruments, New Castle, DE, USA) in the argon atmosphere. Thermograms were recorded in the temperature range of 0–100 °C at a heating/cooling rate of 2 °C/min.

The measurements of the thermal conductivity of paraffin, asphaltenes, and composites were carried out at 20 °C using a KITT-Nanocomposite device (KB “Teplofon”, Russia). Samples were preliminarily prepared from paraffin or composites in the form of tablets 10 mm in diameter and 5 mm thick. Asphaltenes tablet samples were prepared by compaction of the asphaltenes powders in a mold at room temperature under light pressure. Then the samples were placed in a measuring cell made of neutral material. The method of measurement consisted in uniformly preheating a sample and then its monotonous cooling from one side using a Peltier element, with calculation of the value of heat conductivity using the temperature difference at the edges of the sample and the value of heat flow.

The research methodology is presented in Figure 10.

## 4. Conclusions

In the present work, inexpensive carbon additives obtained from wastes of ethylene production by the pyrolysis of hydrocarbon feedstock were studied as enhancers of the thermal conductivity of PCMs. Technogenic asphaltenes and their modified derivatives have been studied as fillers to enhance the thermal conductivity of paraffin. The following conclusions were made:(1)Due to the presence of a polycyclic aromatic core, technogenic and petroleum asphaltenes can be considered as carbon nanofillers, similar in structural properties to small fragments of graphene. The revealed features of the structure of technogenic asphaltenes—high condensation and an almost complete absence of heteroatoms in polyaromatic systems—make them a more promising basis for obtaining PCM thermal conductivity enhancers by chemical modification.(2)The oxidative modification of technogenic asphaltenes makes it possible to change their composition in a controlled way. In the case of oxidation with ammonium persulfate in sulfuric acid, significant changes are observed in the structure of modified technogenic asphaltenes associated with the formation of more condensed structures and a noticeable increase in the oxygen content in the average molecule due to the formation of carboxyl, ether, and sulfo groups, as well as the appearance of a significant amount of heteroatoms, especially on the surface. The process of the oxidation of technogenic asphaltenes with ammonium persulfate in acetic acid does not lead to significant changes in their structure and is associated mainly with the formation of carboxyl and ether groups.(3)It was found that the introduction of 5 and 30 wt. % of such fillers leads to the formation of an internal network structure, which contributes to an increase in the sedimentation stability of their suspensions in paraffin, while intermediate concentrations of asphaltenes are not so effective.(4)As a result, the addition of modified technogenic asphaltenes oxidized with ammonium persulfate in acetic acid in an amount of 5–30 wt% allows the increase of the thermal conductivity of paraffin by 58–72%, respectively.(5)The addition of technogenic asphaltenes oxidized with ammonium persulfate in sulfuric acid does not provide a similar increase in the thermal conductivity of paraffin. This is probably due to the presence of sulfo groups, RSO_3_H, formed in presence of sulfuric acid, which enhance interactions between molecules, but also lead to poor (non-uniform) distribution in the wax. As a result, a significant increase in the thermal conductivity was not achieved.

## Figures and Tables

**Figure 1 molecules-28-00949-f001:**
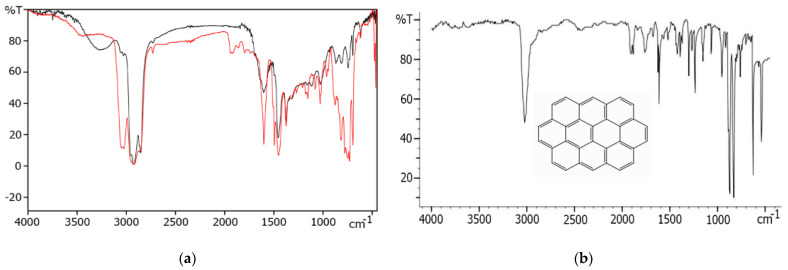
IR spectra of petroleum asphaltenes Petroleum-A (black) and technogenic asphaltenes ET-A (red) (**a**), ovalene (**b**).

**Figure 2 molecules-28-00949-f002:**
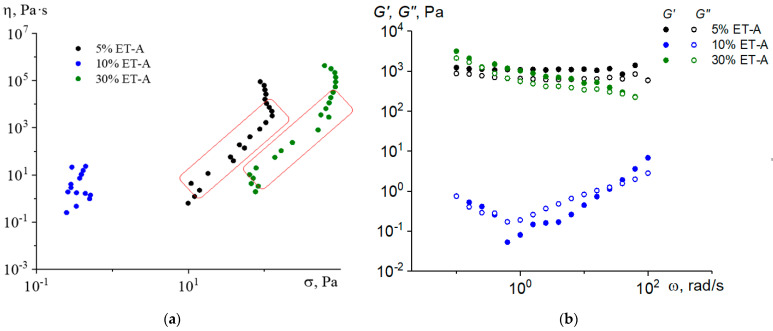
Dependence of viscosity on shear stress (**a**) and frequency dependences of storage and loss moduli (**b**) of paraffin composites containing ET-A asphaltenes, T = 80 °C. Marked areas in (**a**) with reverse slope correspond to deterioration of the structural network of composites.

**Figure 3 molecules-28-00949-f003:**
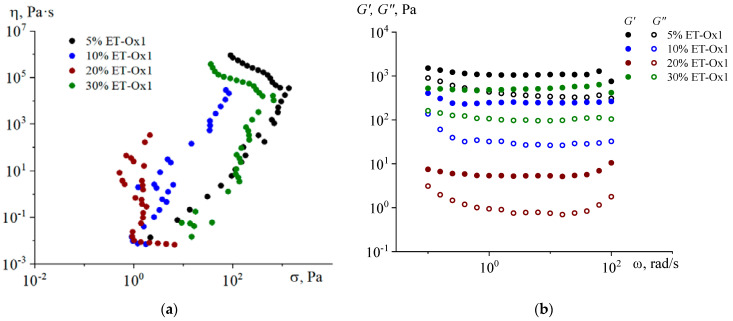
Dependence of viscosity on shear stress (**a**) and frequency dependences of storage and loss moduli (**b**) of paraffin composites containing ET-Ox1 asphaltenes, T = 80 °C.

**Figure 4 molecules-28-00949-f004:**
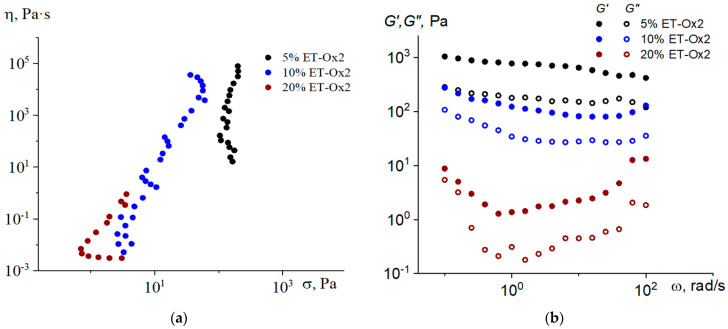
Dependence of viscosity on shear stress (**a**) and frequency dependences of storage and loss moduli (**b**) of paraffin composites containing ET-Ox2 asphaltenes, T = 80 °C.

**Figure 5 molecules-28-00949-f005:**
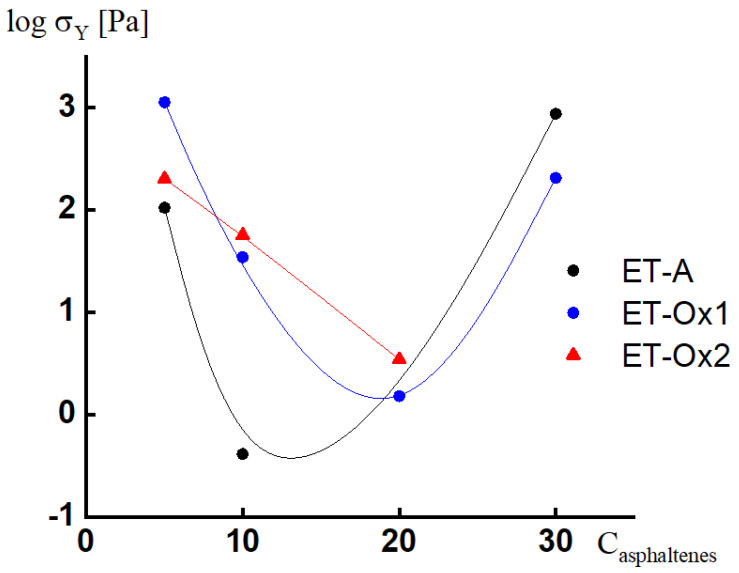
Concentration dependence of the yield stress of paraffin dispersions containing fillers.

**Figure 6 molecules-28-00949-f006:**
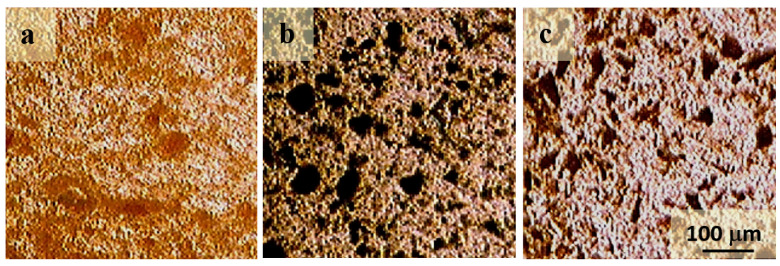
Optical micrographs of paraffin containing 5 wt. % asphaltenes ET-A (**a**), ET-Ox1 (**b**), ET-Ox2 (**c**).

**Figure 7 molecules-28-00949-f007:**
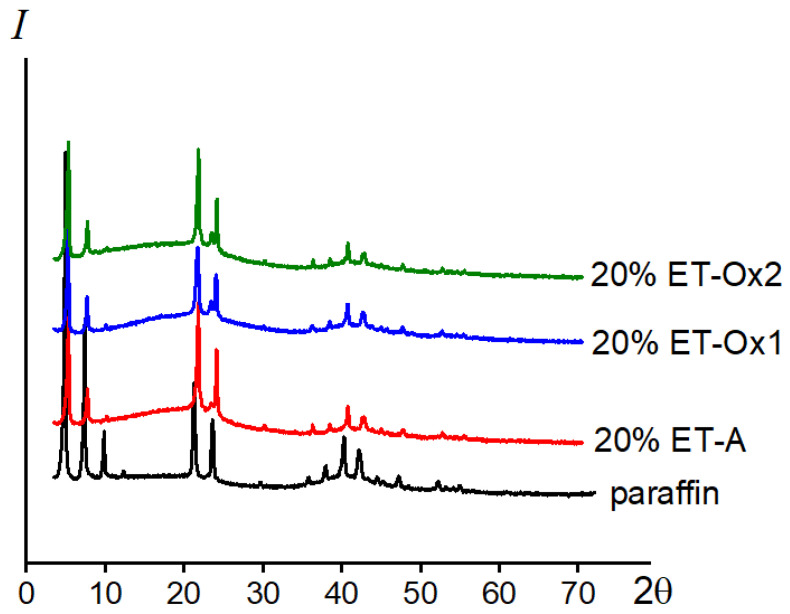
X-ray diffraction patterns of systems based on paraffin (1) and its composites containing 20 wt. % asphaltenes ET-A (2), ET-Ox1 (3), ET-Ox2 (4), T = 25 °C.

**Figure 8 molecules-28-00949-f008:**
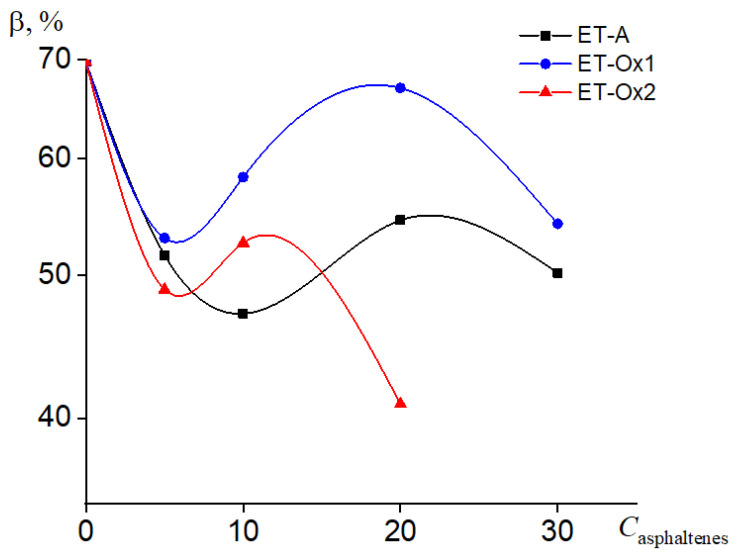
Concentration dependence of the degree of crystallinity of composites based on paraffin containing asphaltenes of various natures.

**Figure 9 molecules-28-00949-f009:**
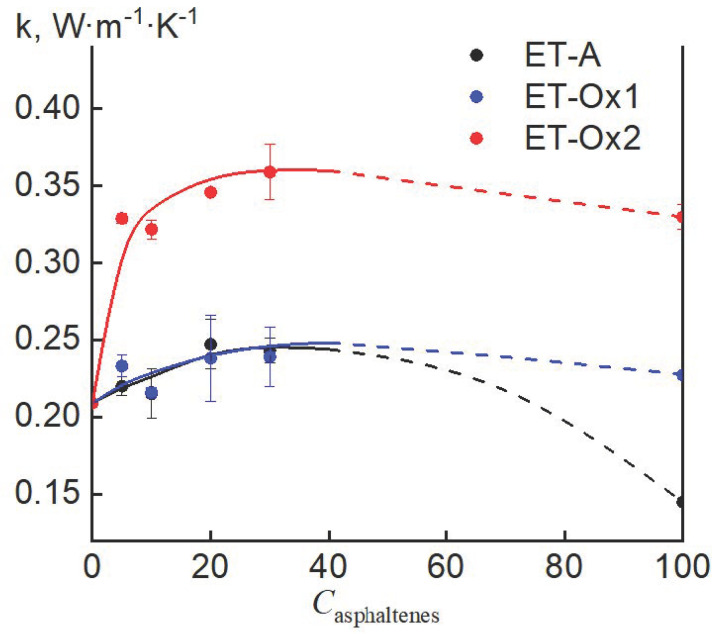
Concentration dependence of the thermal conductivity of composites based on paraffin containing asphaltenes of various nature, T = 25 °C.

**Figure 10 molecules-28-00949-f010:**
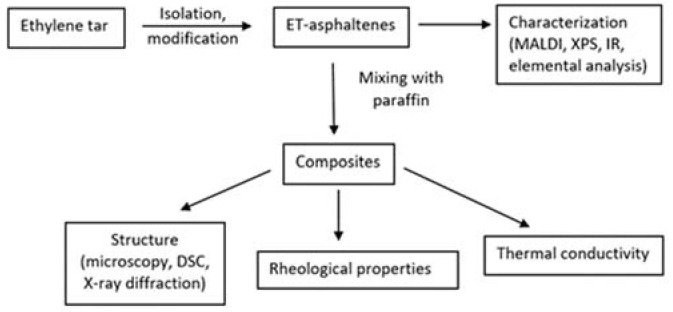
Research methodology flow chart.

**Table 1 molecules-28-00949-t001:** Content of elements in asphaltene samples according to X-ray photoelectron spectroscopy (XPS) data (on the surface) and CHNS elemental analysis.

Asphaltene Samples	Concentration of Elements, wt. %								
	at the Surface (XPS Data)				in the Whole Sample (CHNS Analysis Data)				
	C	N	O	S	C	H	N	S	O ^1^
Petroleum-A	93.0	1.0	2.1	3.6	80.9	8.1	2.1	6.9	2.0
ET-A	99.3	-	0.7	-	92.2	6.5	-	-	1.3
ET-Ox1	71.6	1.1	23.3	4.0	51.9	4.9	1.6	6.4	35.2
ET-Ox2	96.5	-	3.5	-	83.8	6.1	-	0.7	9.4

^1^ the oxygen concentration was calculated as a difference between 100% and the sum of the concentrations of the other elements.

**Table 2 molecules-28-00949-t002:** Spectral coefficients derived from IR spectroscopy data for samples.

Asphaltene Samples	Spectral Coefficients					
	CH_2_/ C=C	C=C/ CH_3_+CH_2_	C=O/ C=C	C-O/ C=C	RSO_3_H/ C=C	S-O/C=C
ET-A	1.33	0.49	-	-	-	-
ET -Ox1	0.82	1.22	0.89	1.20	1.21	1.31
ET -Ox2	1.02	0.54	0.78	0.89	0.68	0.65

## Data Availability

Not applicable.

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
