# Peer review of "Modified Technogenic Asphaltenes as Enhancers of the Thermal Conductivity of Paraffin"

_molecules, 2023, doi:10.3390/molecules28030949_

Round 1

Reviewer 1 Report

In this study technogenic asphaltenes and their modified derivatives were studied as fillers  to enhance the thermal conductivity of paraffin. Detailed experimental results have been obtained and analyzed. However, the paper is recommended to be revised before publication.

1.        Other inexpensive carbon fillers have also been studied in the literature. The present work should be compared with those reported in the literature to demonstrate the novelty.

2.        The reasons for the different effects of the technogenic asphaltenes and their modified derivatives should be discussed more in detail. For instance, why the addition of technogenic asphaltenes oxidized with ammonium persulfate in sulfuric acid does not provide a similar increase in the thermal conductivity of paraffin?

3.        The mechanism behind the phenomena should be discussed. For instance, why at asphaltenes concentrations of 5 and 30 wt% their particles or aggregates form a stable structural network capable of preventing their sedimentation in the paraffin melt while at intermediate concentrations the network is much weaker?

Reviewer 2 Report

As an experimental study, three asphaltenes samples as ET-A, ET-Ox1 and ET-Ox2 have been considered to determine the effect of each one on the Paraffin thermal conductivity. The authors have led to 72% thermal conductivity enhancement for Paraffin in the highest concentration. So, it can be significant in thermal aspect of Paraffin in the phase change processes.

Generally, the manuscript has a suitable technical structure; however, it is necessary to provide some improvements as follow. Then, after removing all the comments, publication of the manuscript in the reference journal can be feasible.   

1. Please make up a nomenclature list and define all the parameters, symbols and abbreviations appeared in the manuscript, or define all variables on first use.

2. Text of the introduction section is very compact. Indeed, there are some reviewed papers which have been shown in an accumulated form. Such format cannot be acceptable. Please provide them stand- lonely as possible as.

3. Please express the proposed novelty of the present study, explicitly. What is the major innovation of the present study against the others in the field?

4. As a common rule in the such papers, the used methods (methodology) should be shown before results and discussion section. The methodology section could be included a flowchart as well. So, please consider it.   

5. Around table 1, in case of it is related to the other reference/s, it is suitable to introduce that reference/s in the caption of table.

6. The mentioned samples in the present manuscript have been called to be inexpensive. The samples should be compared with the common nanoparticles from the economic point view. It can declare using of the samples in the different conditions against the routine nanoparticles or any other additive. Please elaborate.

7. Regarding to the prior comment, please determine overall advantage/ disadvantage of the mentioned samples against regular nanoparticles or hybrid ones.  

8. Regarding the conclusion section, if possible, please provide the main results of the present study in the case/ point format.
